# Building-Information-Modeling Based Approach to Simulate Strategic Location of Shelter in Place and Its Strengthening Method

**DOI:** 10.3390/ma14133456

**Published:** 2021-06-22

**Authors:** Young-Hwi Kim, Jin-Seok Choi, Tian-Feng Yuan, Young-Soo Yoon

**Affiliations:** School of Civil, Environmental and Architectural Engineering, Korea University, 125 Anam-ro, Seongbuk-gu, Seoul 02841, Korea; younghwi@korea.ac.kr (Y.-H.K.); radiance@korea.ac.kr (J.-S.C.)

**Keywords:** building information modeling, evacuation simulation, shelter in place, low-velocity impact, strengthening method

## Abstract

It is important to consider establishing a shelter in place (SIP) using existing facilities to prepare for unpredictable and no-notice disasters. In this study, we evaluate the building-information-modeling (BIM)-based approach to simulate the strategic location of SIP and its strengthening method. BIM software was used to model a light rail station and analyze the elements of the facility that can affect the evacuation time to reach the SIP. The purpose of this study was to understand the effects of structural standards on the design of SIPs using a direct simulation. The differences between domestic and overseas standards were analyzed. An analysis was carried out to evaluate whether national specifications are satisfactory. As the proposed evacuation method is based on a rational human behavior analysis through a direct simulation, it was going to be a safer and faster route of evacuation in the case of physical terror attack situations for existing infrastructure, Furthermore, the SIP design is considered where reinforcement of the SIP structure is necessary. Three types of reinforcing were considered. Here, the use of high-strength, high-ductility concrete proved to be an effective method to improve the impact resistance of reinforced concrete walls and recommended for strengthening reinforced concrete members.

## 1. Introduction

Protective action decisions are an important emergency planning issue for public facilities. The decision whether evacuation or shelter in place (SIP) should be recommended is one of the most important questions for local emergency managers in response to critical situations [1,2,3,4].

However, the validation of facilities that affect the evacuation time during the design of the SIP has not been extensively researched. The design of SIPs in existing facilities would be preferable to a random increase in the number of evacuation facilities, and those facility would be utilized for general purposes in normal circumstances.

SIP is suggested for the first time in Israel to protect civilians. Nowadays, this is the concept of enhanced evacuation proposed by the United States. Based on the various advantage (accessibility, maintenance, repair) of SIP, it is the best choice for Korea as an evacuation facility, which has a high population density.

The simulation of the strategic location of SIP in the facility generally adopts the roadmaps for building information modeling (BIM), which has a three-dimensional (3D) object design paradigm. The BIM roadmap was first used in 2010 in the Public Procurement Service (PPS), where they announced it for all facility projects [5,6]. Therefore, a BIM model of a light rail station for public facility was established. This study aims to identify the facility elements that should be considered for the design of SIP, and then confirm whether the Korean specifications of those are satisfactory. This study is different from previous research, in which 3D modeling was established using BIM software. The facilities were confirmed by analyzing the location, accessibility, and user’s movement according to evacuation simulation.

In this study, an underground light rail station, which has the characteristics of both building and civil infrastructure, was adopted as a sample model and modeled using the Autodesk Revit software. The evacuation simulation was using the pathfinder, which developed by Thunderhead Engineering [7].

Hence, this paper focuses on the BIM-based approach to simulate the strategic location of SIP and its strengthening method. When the SIP design is considered, reinforcement of the SIP structure is necessary. In the last several decades, researchers have developed various materials and techniques for strengthening reinforced concrete (RC) structures. Among the most widely used is the carbon fiber reinforced polymer (CFRP) sheets and metal grid that are more flexible. However, strengthening with CFRP does not perform effectively in compression and bond properties under loading. A more recent material developed and used for both repair and strengthening of RC members is the high-performance fiber reinforced concrete. Therefore, a new material named high-strength, high ductility (HSDC) has been used in this study. The materials characteristics of HSDC were evaluated to validate the effectiveness of the mixture design approach and prove the possibility of combining both high strength (over 120 MPa) and ductility (tensile strain over 2.9%) into concrete without heat treatment. HSDC showed approximately 4.4 times and 76% greater than those of typical ultra-high-performance concrete in the tensile strain capacity and toughness, respectively. Further, HSDC showed approximately 0.76 higher tensile strength, and 0.53 lower tensile strain capacity than typical engineering cementitious composites (Section 4.3.1).

Thus, the four types of RC walls (non-strengthened specimen; top and bottom strengthened with HSDC; top and bottom strengthened with HSDC and CFRP, respectively; top and bottom strengthened with metal grid) are evaluated using low-velocity impact load in this study.

## 2. SIP Simulation

### 2.1. Variables for the Evacuation Simulation

#### 2.1.1. Movement of Occupants

Gait speed is closely related not only to the physical condition of the pedestrian, but also to the individual’s habits and personality. In addition, it is very difficult to generalize and define because it is also related to gender, age, birth city, and parental disposition. The maximum unhindered travel speeds to be used are those derived from published data, which provide male and female walk rates as functions of age [8]. The difference in gait speed between women and men is shown in Figure 1, and the function for the difference in gait speed between men and women by age is shown in Table 1. As shown in Figure 1 and Table 1, walking speed is affected by age and gender [8,9]. In addition, structural shapes such as tread, height, width of stair, and door width can also affect gait speed. [10,11,12].

Since various factors affect gait speed, it is difficult to define gait speed considering all factors. Therefore, in this study, the gait speed was defined according to age and gender, which had the greatest influence on the gait speed and was judged to be suitable for generalization. In addition, since the standard according to the stair shape information must be designed/constructed based on the codes of each country, in this study, the horizontal speed was applied based on the gait speed according to age/gender, and the vertical speed was mainly different from the horizontal speed. There is a bottleneck, which is greatly affected by density. Therefore, in this study, it is applied according to the density as defined by the SFPE (Society of Fire Protection Engineers) (Figure 2) [13]. The horizontal speed of occupants applied in this study was referred to by Richard et al. [14], and the gait speed under the age of 20, which is not defined here, was calculated based on the previous paper [8].

Pathfinder supports two-movement simulation modes. In “Steering” mode, occupants use a steering system to move and interact with others. This mode tries to emulate human behavior and movement as much as possible. SFPE mode uses a set of assumptions and hand-calculations as defined in the Engineering Guide to Human Behavior in Fire (SFPE, 2003). In SFPE mode, occupants do not attempt to avoid one another and are allowed to interpenetrate, but doors impose a flow limit and velocity is controlled by density [7]. Figure 3 visually expresses how the two pedestrian modes differ by indicating pedestrian movement. Therefore, in this study, steering mode is selected to perform in a more realistic.

#### 2.1.2. Occupants Counting for Evacuation Simulation

To simulate evacuation time according to the location of the SIP, there are many variables necessary for the simulation. Among them, the calculation of the number of people evacuated is the variable with the highest priority. In this study, since the light rail station was sampled as a model, it was decided to simulate with passengers using the station. Further, in Section 2.1.1. Movement of Occupants, among the factors affecting the evacuation speed, gender and age are important, so when calculating the number of passengers, it should be categorized by age and gender. To this end, referring to the 2017 National Statistical Office data [15], among the total population of Korea, there were 24,922,000 men and 25,021,000 women, which was found to be 98,000 more women than men. In this study, light rail users were classified by age and gender [16,17].

The longest stay at the station is 5.708 min, considering that it takes 2.708 min for subway users to enter the station and get to the platform, and the train dispatch interval is up to 3 min during the station’s most complex commute. Conversely, the time taken from the platform to the exit to go outside the station after using the subway was simulated as 3.132 min [16]. Therefore, the time that subway passengers spend the most at the station is 3.132 min, which was taken into consideration and the number of simulations was calculated (Table 2).

#### 2.1.3. Location of Shelter in Place

To select the location of the SIP, it is necessary to check the walking movement of pedestrians (Figure 4). The light rail station used in this study is an underground facility, and in order to move to the platform on the 4th basement floor, pedestrians who enter from the ground can descend from the ground to the 1st basement and then use the escalator, stairs or elevator to move to the 4th basement floor. The 2nd and 3rd basement floors do not overlap with pedestrian traffic and are used by station staff. Therefore, the 2nd and 3rd basement floors are the most suitable for evacuation facilities in terms of location and environment under the CBRE situation. As a result of examining the capacity of the two floors (Figure 5), the area of 2nd basement floor which can use for SIP was 356,242 m^2^ and the 3rd basement floor which can use for SIP was 536,285 m^2^ (Table 3). FEMA 453 limited the duration of occupancy to more than 24 h only for wartime [18]. Therefore, if only the case of less than 24 h is considered, the 3rd basement floor is suitable, but the case where both the 2nd and 3rd basement floors are considered was reviewed in addition to the simulation.

### 2.2. Modeling

#### 2.2.1. Creation of a Mesh

The main egress components include rooms, which are empty floor spaces bounded by walls, doors, which connect rooms on the same level, stairs/ramps, which connect rooms on different levels, and elevators, which connect multiple levels. To create each component, the most basic modeling element is the Mesh, and the mesh is connected by points and lines. Management of the 3D triangulated mesh is important to define the width of the door, the width of the stairs, the height of the stairs, and the connection between objects.

#### 2.2.2. Door

Since the entrance is narrow, it causes a bottleneck when occupants gather, so it has the greatest influence on the evacuation time. Since the simulation time it takes for a pedestrian to reach a specific location is an important factor to consider when selecting a SIP location, it is important to define the door according to the conditions when modeling. The variable of door has width, height, and depth, and to define each variable, it is possible only when the mesh is precisely created at the location to be defined. The number of pedestrians passing through the door can be checked, and this can be used as a reference in the design of the door width and SIP circulation to reduce the bottleneck.

#### 2.2.3. Escalator and Stairs

The factors that have a big influence on the evacuation simulation in terms of shape can be stairs or escalators like doors. A bottleneck occurs when there are many occupants entering a narrow staircase or escalator from a wide room or corridor. At this time, the gait speed may be affected according to the height, width, and tread shape information of the stairs. Unlike a stationary staircase, an escalator is a moving facility, and there are two ways to model it in Pathfinder. After modeling the same as stairs, there is a way to define the direction of the escalator as one way. After modeling as a ramp, you can also set the escalator by marking +x and -x directions. The escalator speed is suggested to be 0.5 m/s to 0.67 m/s in the design standard for a subway (Facility Field) [19], and in this study, it was set to 0.5 m/s conservatively.

#### 2.2.4. Elevator

Elevators in subway stations are not for ordinary citizens who use the station, but for people with reduced mobility, such as the disabled, the elderly, and pregnant women. Unlike elevators installed in buildings or public facilities, the elevators are slow to ascend and descend. The elevator speed is set at 0.75–1.0 m/s according to the elevator installation standard of the design standard for subway [19].

#### 2.2.5. Occupants

As discussed in Section 2.1.1, the number of people was automatically calculated according to age and gender, and the pedestrian speed was applied accordingly [16,17]. Of the 213 simulated personnel calculated in Section 2.1.2, 208, excluding five station attendants, were on the 1st basement floor and four basement levels. They were evenly distributed on the platform on the first floor, and three out of five station staff were placed on the 1st basement floor and the other two on the 4th basement platform.

The calculated occupants were equally applied to all scenarios and to simulate the effect on the disabled. The disabled accounted for 13.8% of the total Korean population. Among the disabled, 33.9% was surveyed as a disabled person in need of help for daily living, equating to 4.67% of the total population. Therefore, out of 213 simulated people, 10 out of 208 excluding station staff, or 4.67%, were calculated as the people who needed help when evacuating. When evacuating, occupants or station staff around the disabled recognize the disabled and help the disabled to move to the SIP using the elevator (Figure 6).

## 3. Strengthening Method for Resistance to Impact Load

### 3.1. Test Specimens

The SIP structures included mainly three types of strengthened RC walls. The mixture proportions used for the fabrication normal concrete (NC) and high-strength high-ductility concrete (HSDC) are summarized in Table 4. The chemical and physical properties of the materials are listed in Table 5. The three RC specimens had dimensions of 1600 mm^3^ × 1600 mm^3^ × 140 mm^3^ and were doubly reinforced with equal amounts of reinforcement (*f_y_* = 400 MPa, *x* direction = D13 @ 240 mm, *y* direction = D13 @ 210 mm) in the top and bottom layers (NC-NN). The other three types of specimens were distinguished by different strengthening configurations: top and bottom sides jacketing HSDC (NC-HS); top sides jacketing HSDC and bottom reinforcing CFRP sheets (NC-HF, 4900 MPa of tensile strength, 230 GPa of elastic modulus, and 0.167 mm of thickness); and top and bottom sides reinforcing with 5 mm metal grid (NC-MG), respectively. The details are shown in Figure 7.

### 3.2. Test Set-Up

The RC specimens were fabricated and tested using a low-velocity drop-weight impact test. The clear span of each specimen in the two-way flexural tests was a 1500 mm. Impact loads were applied at the center of the specimens using a weight of 300 kg and drop height of 2000 mm. To measure the maximum and residual deflections, a potentiometer manufactured by COPAL Electronics was located at the center of the specimen, with a capacity of 100 mm. Four load cells were mounted on the supports. Each load cell had a capacity of 50 tons. The potentiometer and load cells were measured using a dynamic data logger (DEWE-43V). The sampling rate of the data logger was 100 kHz, similar to that in recent studies [20].

## 4. Results and Discussion

### 4.1. SIP Simulation According to Various Scenarios

As shown in Table 6, the simulation was performed by dividing it into four scenarios. All scenarios were carried out assuming that occupants for less than 24 h were accommodated without considering sleep in the CBRE (chemical, biological, radiological, and high-yield explosive) situation. In the case of scenario, only the 3rd basement floor is designated as SIP, and in scenarios B~D, both the 2nd and 3rd basement floors are defined as SIP. In the existing design, the only way to go to the 3rd basement floor is to use the emergency stairs from the 1st basement floor. Since this causes concentration of occupants, scenarios C and D were also simulated to install additional emergency stairs from the 4th basement to the 3rd floor.

#### 4.1.1. Scenario A

The 3rd basement floor is designated as SIP in Scenario A. As a result of the simulation, the minimum time to reach the third basement floor was 47.4 s, the maximum time was 396.6 s, and the average time was 218.7 s. The minimum, maximum, and average distances traveled by occupants were 14.0, 189.3, and 105.9 m, respectively. In addition, after 20 s elapsed during the simulation, people gathered intensively to the emergency stairs going down to the 3rd basement which is designated the SIP location in this scenario from the 1st basement. There was a bottleneck in which occupants overlapped in rooms 95 and 135; refer to Figure 8a. As shown in Figure 8b, when 150 s of simulation time elapsed, it was confirmed that a serious bottleneck occurred as a maximum of 10 people gathered in Room 135.

#### 4.1.2. Scenario B

In Scenario B, both the 2nd and 3rd basement floors were designated as SIPs and simulations were performed. Compared to the scale, the 2nd basement floor is smaller than the 3rd floor, so 192 people move to the 2nd floor, and the remaining 21 people move to the 3rd floor. As a result, the simulation time was confirmed as the minimum time for occupants to move to the SIP of 46.6 s, the maximum time of 389.9 s, and the average time of 187.5 s. The minimum, maximum, and average distances traveled by occupants were 15.2 m, 178.1 m, and 90.6 m, respectively. Like Scenario A, in Scenario B, bottlenecks occurred in Room 96 and Room 135 for moving to the emergency stairs, but as the maximum distance required for evacuation was shortened by 5.92% from 189.3 m to 178.1 m, the time taken to evacuate from 397.5 s to 390.0 s, showing an improvement of 1.66%.

#### 4.1.3. Scenario C

In scenarios A and B, for pedestrians to move to the SIP, a path was simulated to move from the first basement to the second and third floors using the emergency stairs. Since this is a structure that inevitably causes bottlenecks in the doors and stairs on the first basement floor, in scenario C, an emergency stair from the 4th basement to the 3rd basement level was additionally installed. In the simulation, 121 pedestrians on the first basement floor and stairs use the emergency stairs on the first basement floor, and 92 people on the platform move to the third basement floor using the emergency stair which is installed additionally. (Figure 9a). As a result, the minimum evacuation time was 21.8 s, the maximum was 291.9 s, and the average was 127.3 s, and the minimum distance traveled by the occupants was 14.4 m, the maximum was 170.1 m, and the average was 76.4 m. Compared with scenario A, the evacuation time was reduced by 98.6 s (24.86%), from 396.6 s for scenario A to 298.0 s for scenario C. In Scenarios A and B, the bottleneck occurred in Room 95 on the first basement floor, but in Scenario C (Figure 9b), the bottleneck was relatively reduced.

#### 4.1.4. Scenario D

In Scenario C, emergency stairs were additionally installed only on the upbound line, whereas in Scenario D, emergency stairs were installed on both the up and down lines. As a result, compared to Scenario C, the maximum evacuation time was reduced by 5.537% from 298.0 s to 281.5 s, and the average evacuation time was reduced from 127.3 s to 120.2 s, reducing the maximum travel distance of pedestrians by 5.58%.

### 4.2. Improvement Method for Evacuation

The next review is to analyze how the evacuation time is affected by the riser and tread of the stair under the same conditions as in scenario A. In the case of riser, it was reviewed as 180 mm in consideration of the maximum height restricted by BRE, IBC, and LSC to reduce the misstep suggested [21], and for the tread, 280 mm or 305 mm suggested by overseas standards is applied like the riser. Case A and B change and review the riser; Case C, D, E change the tread; and Case F changes the stair width and analyze the overall result (Table 7).

#### 4.2.1. Case A

In Case A, the previously installed stair riser was 158.2 mm–176.5 mm, but after changing all of them to 180 mm, what effect it has on the evacuation time was reviewed. As a result, the maximum time for occupants to move to the 3rd basement floor, which is the SIP location, increased by 6.9 s (1.74%) from 396.6 s to 403.5 s, and the evacuation distance also increased by 4.5 m (2.38%) from the maximum distance of 189.3 m to 193.8 m. Since the riser is 200 mm or less [21], both 180 mm and 158.2 mm–176.5 mm meet the design code, but it was confirmed that the evacuation time and evacuation distance were increased when the stair riser was higher than the existing stair riser.

#### 4.2.2. Case B

In Case A, the riser of all stairs was set to 180 mm. In Case B, the result was checked when the maximum riser height suggested [16,17,21], 200 mm, was applied under the same conditions. As a result of increasing the height of the existing riser by 21.8 mm from 158.2 mm to 176.5 mm, the evacuation time increased by 10.1 s (2.55%) from 396.6 s to 406.7 s, and the evacuation distance increased by 11.8 m (6.23%) from 189.3 m to 201.1 m. As confirmed in Cases A and B, when the riser height is high, evacuation time and movement time increase, which is unfavorable for evacuation.

#### 4.2.3. Case C

Case A and B reviewed the riser of the stair, but in Case C, how it affects evacuation by changing the stair tread. Existing tread is 280 mm–300 mm, but after changing all of these to 300 mm, the simulation result confirmed that the evacuation time was reduced by 3.0 s (0.76%) compared to the existing tread to a maximum of 393.6 s, and the maximum travel distance taken in the evacuation time increased by 4.9 m (4.59%). Although the tread was partially lengthened, the travel distance was partially increased, but it was confirmed that it worked advantageously for the evacuation time.

#### 4.2.4. Case D

In Case C, all stair treads were changed to 300 mm and the results were checked, but in Case D, all of them were 350 mm, which had an added 50 mm. As a result, the maximum travel time required for occupants to evacuate was 374.6 s and the maximum travel time was 201.0 m. Similar to Case C, the maximum travel distance increased by 11.7 m (6.18%), but the evacuation time was shortened by 22 s (5.55%).

#### 4.2.5. Case E

In Case D, the tread was increased to 350 mm. In Case E, the tread was increased by 50 mm and set to 400 mm, and the result was checked. As a result of the simulation, it was confirmed that the maximum evacuation time was 372.8 s and the maximum travel distance was 192.7 m, which was 3.4 m (1.80%) longer than before, and the evacuation time was shortened to 23.8 s (6.0%).

#### 4.2.6. Case F

In Case F, if the width of the stair rather than the tread and riser of the stair is changed, the effect it has on the evacuation simulation result was reviewed. In this scenario, the width of the emergency stairs at this station is partially changed from 1000 mm to 1200 mm, and the simulation results are checked. As a result of the simulation, the maximum evacuation time was 361.6 s, and the movement distance was 195.3 s. With this tread change, the evacuation movement distance increased by 6 m (3.17%), and the required time was shortened to 35 s (8.83%).

### 4.3. Results of the Low-Velocity Impact Test

#### 4.3.1. Properties of Strengthening Materials

After the BIM-based approach to simulate the strategic location of SIP, the structure (concrete walls) was strengthened by three types of materials (CFRP sheets, metal grid, and HSDC). The CFRP sheets and metal grid were used off the shelf; however, HSDC was a new material researched to use for strengthening.

HSDC is mixed with 1.5% fiber volume fraction of polyethylene fiber (12 mm length, 387 aspect ratio, and 0.5 vol.%) and steel fiber (19.5 mm length, 97.5 aspect ratio, and 1.0 vol.%). The compressive strength (ASTM C39), flexural strength (ASTM C1609), and tensile strength (JSCE) occurred as 123.4 MPa, 21.9 MPa, and 9.7 MPa without heat treatment on the 28th day, respectively (Table 4). It should be noted that the post-crack tensile strain was exhibited as 2.78%. The tensile strength of the HSDC was approximately 33.0–39.2% lower than that of typical ultra-high-performance concrete, which uses different lengths of high-strength straight steel fiber and 1.5 vol.% or 2.0 vol.% curing with heat treatment. However, the tensile strain capacity and toughness of HSDC were 4.4 times, and 0.76 higher than those of typical ultra-high-performance concrete, respectively [22,23]. Compared with typical engineered cementitious composite [24], the HSDC showed approximately 63.3–76.0% higher tensile strength, 53.7–54.5% lower tensile strain capacity, and 0.24–0.32 lower toughness (Figure 10). Overall, the HSDC without heat treatment exhibited perfect tensile properties compared with those of previous mixtures. Furthermore, bond strength of HSDC to concrete was evaluated with slant shear tests, conducted according to ASTM C882. The specimens with HSDC conformed to the properties (14–21MPa for 28 days) of the ACI Committee 546 recommendation. Six reinforcement concrete beams (strengthening by HSDC) were prepared and tested according to a three-point bending test, which was evaluated the flexural capacity of normal concrete beams strengthened with HSDC [25,26].

The HSDC exhibited great mechanical properties, tensile strain capacity, energy absorbing, and positive synergy compared to the other cementitious materials [25,26]. Therefore, the reinforced concrete wall strengthened with HSDC was evaluated and compared with CFRP and metal grid.

#### 4.3.2. Damage Level and Failure Mode

Support rotation is used as an important response parameter in the design criteria, defined as the ratio of the maximum displacement to the middle span length recommended in UFC-3-340-02 [27]. This parameter is relatively simple to measure in the low-velocity impact test, in which the deflection-time history of the test specimen is measured and used to calculate the support rotation values.

The corresponding damage level was then evaluated according to the design criteria provided by UFC-3-340-02. The protective design manual defines the base of support rotation at three levels, light damage (0° ≤ θ < 2°), moderate damage (2° ≤ θ < 5°), and severe damage (5° ≤ θ < 12°).

Table 8 shows the damage level evaluated using the conventional approach according to the support rotation. All specimens were in the range of moderate damage in the first impact blow based on the support rotation values in the range of 2 to 5 degrees. The specimen reinforced with HSDC exhibited a low support rotation compared to the other specimens in the same impact blow, because of the strengthening by HSDC as well as improved structural properties (such as rigidity and energy distribution capacity) and nearer to over-reinforced RC structure elements [25,26,27,28,29,30,31]. Hence, NC-HS failed in the third impact blow without bond failure, which shows better impact resistance properties than the others. The local shear failure occurred in the third impact blow. The support rotation value was similar to that of the second blow. It is because HSDC has great tensile properties (tensile strength and strain capacity), energy absorbing capacity [26], and bond strength with concrete [26]. The NC-HF specimen exhibited CFRP bond failure in the second impact blow. Thus, the exact maximum displacement value could not be measured. This is a typical problem in RC structures strengthened using CFRP, as in previous studies, (problems with bond strength, durability, ductility, and installation problems) [25,32,33]. For the NC-MG, the local shear failure occurred in the second impact blow and the highest value of the support rotation value was observed, compared to the other specimens. The crack scenario and failure mode of the specimens after the final impact are shown in Figure 11.

#### 4.3.3. Reaction Force

Figure 12 shows the maximum reaction force measured for each impact blow. The maximum reaction force decreased as the impact decreased. In the NC-NN, and NC-MG specimens, local shear and scabbing failure occurred at the first and second impact blows, respectively. In contrast, NC-HF exhibited bond failure and scabbing failure during the second loading blow. The specimen strengthened by HSDC exhibited 11.0–40.9% higher reaction force in the first blow than those of the other specimens, because of the strengthening by HSDC as well as improved structural properties and nearer to over-reinforced RC structure elements [25,26,27,28,29,30,31]. In addition, owing to the bridging effect of the fiber reinforcement, the damaged area at failure was significantly decreased. Although a tendency for scabbing was observed, the fragmentation largely decreased.

Assuming that the kinetic energy of the impact mass is transferred to the specimens without any loss, the total imparted energy until failure of each specimen can be obtained, as shown in Figure 13. Although it varies depending on the loading history in the experimental program, the total imparted energy can be a useful factor for comparison of the specimens tested in this program. The total imparted energy of the NC-HS and NC-HF specimens was 1.2 to 3.2 times higher than that of the NC-NN specimen. This indicates that the dense reinforcement layout and fiber reinforcement significantly improved the impact capacity.

## 5. Conclusions


(1)The light rail station used in this study was designed without considering the location and movement of the SIP. However, when the SIP location and evacuation route were taken into consideration (Scenario B), the evacuation time was reduced by 6.6 s (1.66%) from 396.6 s to 390 s. In the case of design considering the evacuation route (Scenarios C and D), the evacuation time was reduced by 115.1 s (29.02%) to 281.5 s. As a result of this study, it was confirmed that the evacuation time and movement distance were greatly affected when designing the SIP in consideration of the location and movement of the SIP.(2)The stair is one of the structures that has a great influence on the evacuation time. Considering the evacuation time, it was reviewed whether the building code [34] was met. Stair shape information can be categorized to riser, tread, and width of stair, and both riser and tread are restricted differently from the building code [34], so values suggested by other overseas standards such as IBC and LSC, NIST, and BRE were considered and reviewed. As a result, it was confirmed that the lower the riser, the longer the tread, and the wider the width, the more positive effect on the evacuation time. Therefore, in the case of riser, the existing building code [34] limits it to 200 mm or less, but it is recommended to change it to 180 mm or less. In the case of tread, it is recommended to change the existing 240 mm or more to 300 mm or more. In the case of width, it varies from 900 mm [35] to 1200 mm [34] or more, and it is recommended to unify this value and change it to 1200 mm.(3)Regarding the impact test results, the specimens with HSDC reinforcement exhibited significantly improved impact resistances, whereas the total imparted energy of the RC specimen strengthened by HSDC appeared to be 1.2–3.2 times higher than that of the other strengthened RC specimens. However, the specimen strengthened by the CFRP sheet encountered sudden destruction of the CFRP at the second blow. Therefore, the RC reinforced by HSDC reinforcement methods exhibited a higher impact resistance.


## Figures and Tables

**Figure 1 materials-14-03456-f001:**
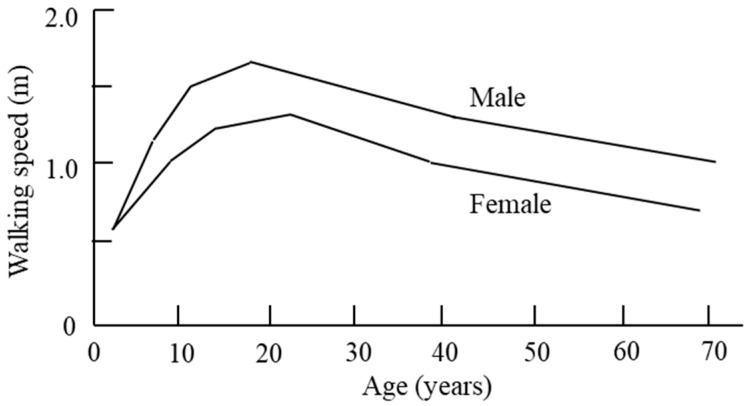
Gait speed graph by age and gender.

**Figure 2 materials-14-03456-f002:**
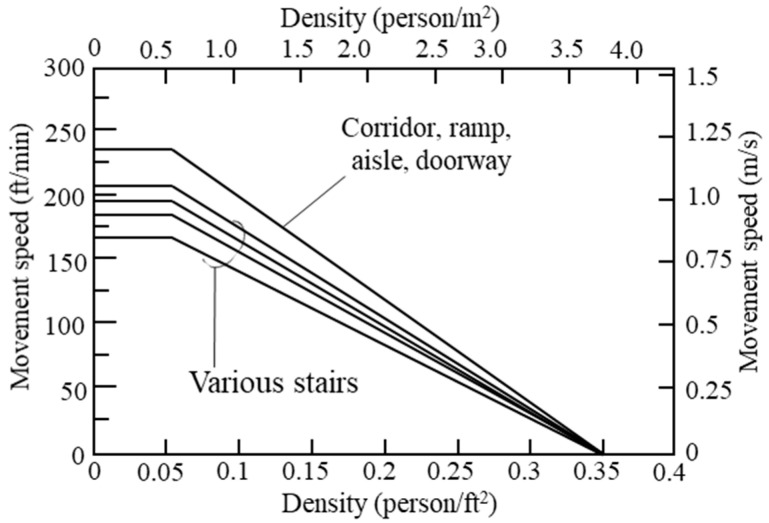
Evacuation speed as a function of density.

**Figure 3 materials-14-03456-f003:**
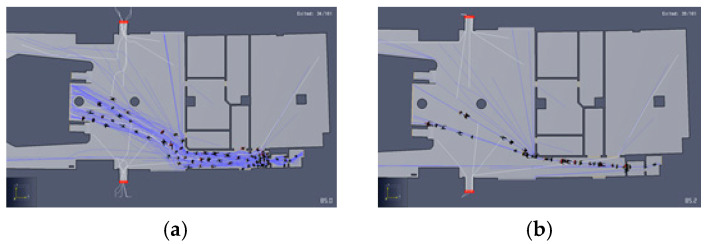
Route results by Simulation Mode: (**a**) steering mode, (**b**) SFPE mode.

**Figure 4 materials-14-03456-f004:**
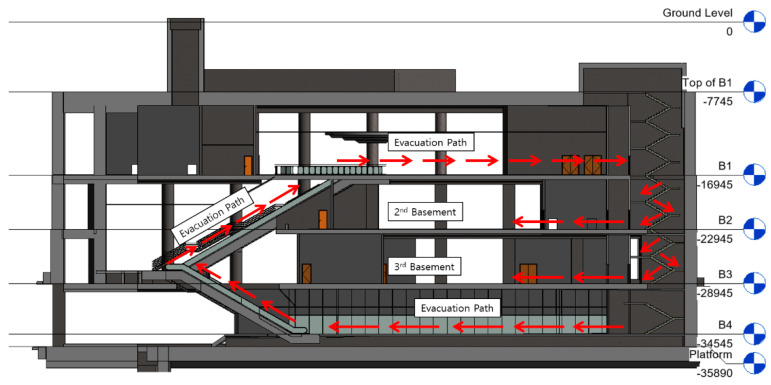
Evacuation path.

**Figure 5 materials-14-03456-f005:**
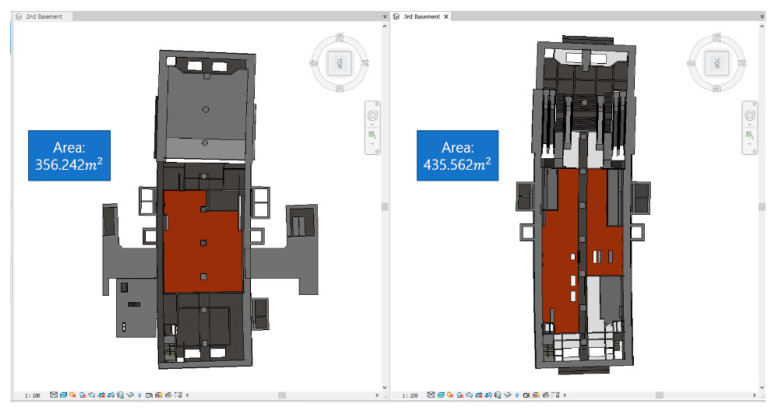
SIP location and areas (the area for the 2nd basement and 3rd basement).

**Figure 6 materials-14-03456-f006:**
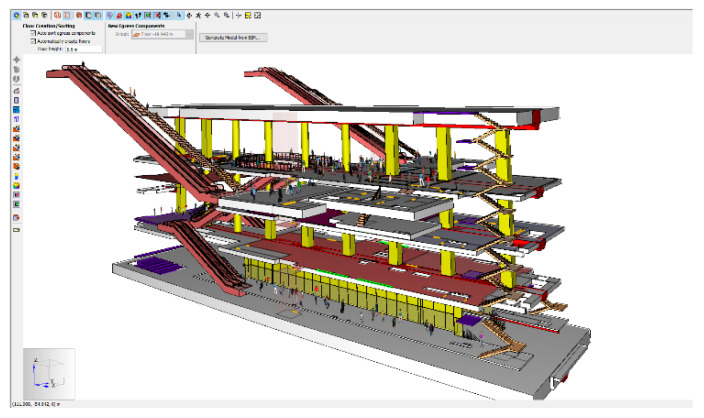
Simulation model with occupants.

**Figure 7 materials-14-03456-f007:**
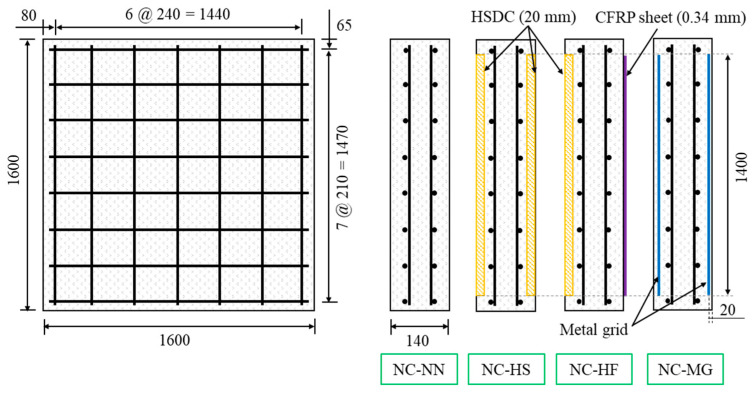
Details of specimens.

**Figure 8 materials-14-03456-f008:**
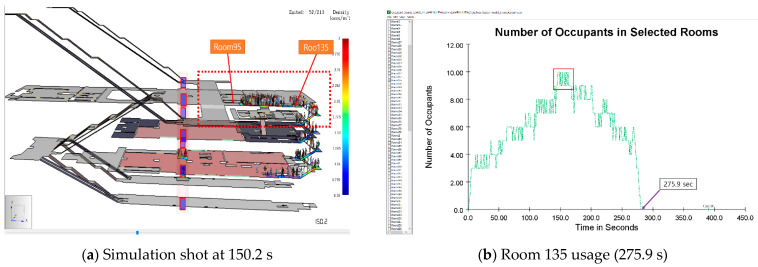
Results of Scenario A.

**Figure 9 materials-14-03456-f009:**
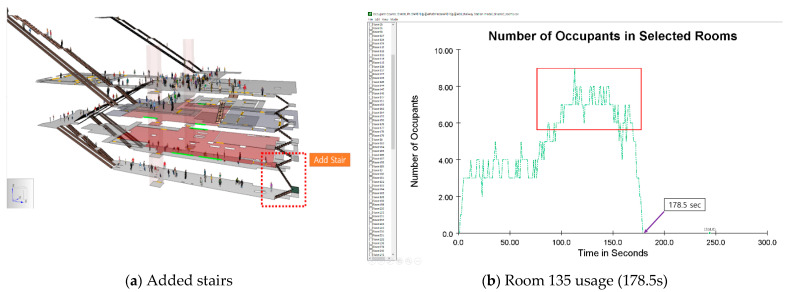
Results of Scenario C.

**Figure 10 materials-14-03456-f010:**
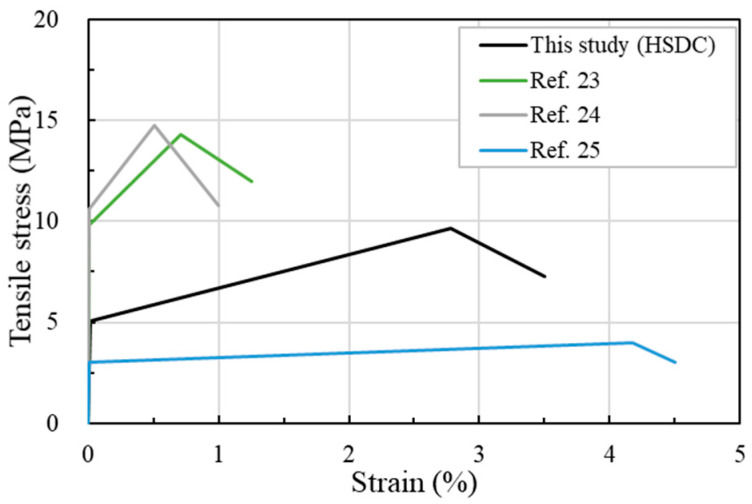
Comparison of tensile stress and strain.

**Figure 11 materials-14-03456-f011:**
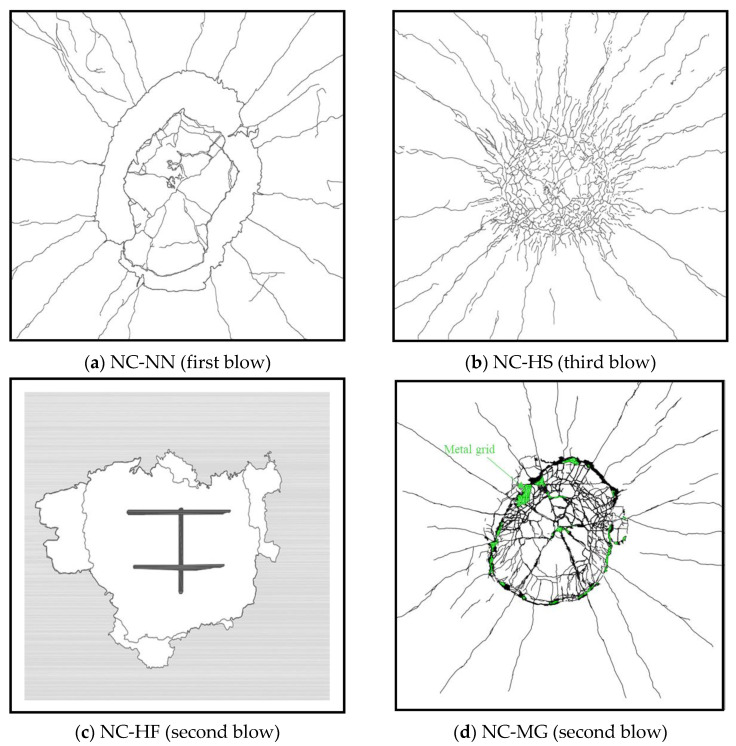
Crack distribution and failure mode.

**Figure 12 materials-14-03456-f012:**
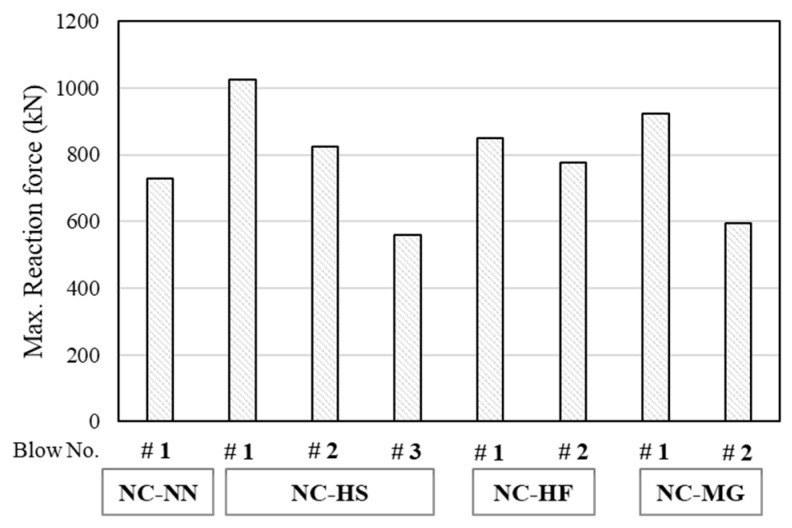
Maximum reaction force.

**Figure 13 materials-14-03456-f013:**
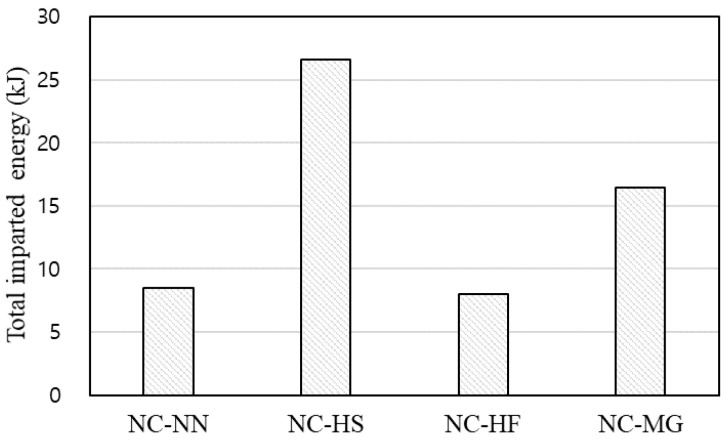
Total imparted energy until failure.

**Table 1 materials-14-03456-t001:** Constants for Equation, Evacuation Speed.

Exit Route Element		*k*_1_ (ft/min)	*k*_2_ (m/s)
Corridor, aisle, ramp, doorway		275	1.40
Riser of stair (mm (in.))	Tread of stair (mm (in.))		
196 (7.5)	254 (10)	196	1.00
272 (7.0)	279 (11)	212	1.08
165 (6.5)	305 (12)	229	1.16
165 (6.5)	330 (13)	242	1.23

Note: S denotes the speed along the line of travel, K denotes constant.

**Table 2 materials-14-03456-t002:** Number of passengers in the station.

Type	Process	Results (Persons)
Calculating occupants to take the train	47 people/min × 3.132 min	147
Calculating occupants to get off	20 people/min × 3.142 min	61
Station attendant	-	5
Total	147 + 61 + 5	219

**Table 3 materials-14-03456-t003:** Shelter capacity.

Type	2nd Basement	3rd Basement
Area (m^2^)	356.24	435.56
Less than 24 h (persons)	192	234

**Table 4 materials-14-03456-t004:** Proportions of the mixtures.

	*w/b*	W	C	FA	CA	SF	F	SS	HSF (%)	PF (%)	SP (%)	*f_ck, 28_* (MPa/COV)	*f_flex, 28_* (MPa/COV)	*f_ten, 28_* (MPa/COV)
NC	0.43	0.43	1.00	2.15	2.42	-	-	-	-	-	0.8	46.2/0.5	26.5/2.4	3.9/3.5 *
HSDC	0.172	0.215	1.00	-	-	0.25	0.30	1.10	1.0	0.5	3.0	123.4/0.3	21.9/5.1	9.7/4.1 **

Note: NC = normal concrete; HSDC = high-strength, high-ductility concrete; *w/b* = water-to-binder ratio; W = water; C = cement; FA = fine aggregate; CA = coarse aggregate; SF = silica fume; F = filler; SS = silica sand; HSF = high strength straight steel fiber (0.2 mm of diameter, 97.5 of aspect ratio); PF = polyethylene fiber (31 μm of diameter, 387 of aspect ratio); SP = superplasticizer, ***f_ck, 28_*** = compressive strength at the 28th-day (ASTM C39); ***f_flex, 28_*** = flexural strength at the 28th-day (ASTM C1609); ***f_ten, 28_*** = tensile strength at 28th-day; * = splitting tensile strength(ASTM C496); ** = direct tensile strength (JSCE).

**Table 5 materials-14-03456-t005:** Physical and chemical properties of materials.

Type	Surface Area (cm^2^/g)	Density (g/cm^3^)	Chemical Composition (%)						
			SiO_2_	Al_2_O_3_	Fe_2_O_3_	CaO	MgO	SO_3_	Na_2_O
C	3492	3.15	21.2	4.7	3.1	62.8	2.8	2.1	-
SF	200,000	2.20	96.0	0.3	0.1	0.4	0.1	<0.2	-
F	2.65	0.75	99.6	0.3	0.03	0.01	0.006	-	0.009

Note: C = cement; SF = silica fume; F = filler.

**Table 6 materials-14-03456-t006:** Location of Shelter in Place and Scenario description.

Scenario	Number of Occupants		Description
	2nd Basement	3rd Basement	
**A**	-	213	Designated 3rd basement floor as SIP
**B**	192	21	Designated 2nd and 3rd basement floors as SIP
**C**	121	92	Add one side of stairs between the 3rd and 4th basement floor (Same SIP location as Scenario B)
**D**	121	92	Add both side of stairs between the 3rd and 4th basement floor (Same SIP location as Scenario B)

**Table 7 materials-14-03456-t007:** Defining Variable Values by Case.

Case	Variable of Stair			Description
	Riser (mm)	Tread (mm)	Stair Width (mm)	
**Origin**	158.2–176.5	280–300	1000	-
**A**	180	280–300	1000	Change riser to 180
**B**	200	280–300	1000	Change riser to 200
**C**	158.2–176.5	300	1000	Change tread to 300
**D**	158.2–176.5	350	1000	Change tread to 350
**E**	158.2–176.5	400	1000	Change tread to 400
**F**	158.2–176.5	280–300	1200	Change stair width to 1200

**Table 8 materials-14-03456-t008:** Location of the SIP and occupant personnel after the modification.

Specimens	Blow No.	Max. Displacement (mm)	*D_max_*/*D_res_*	Support Rotation (Degree)	Damage Level (UFC-3-340-02)
NC-NN	1	23.4	3.02	3.5	Moderate
NC-HS	1	18.7	2.93	2.9	Moderate
	2	27.6	1.96	5.2	Severe
	3	20.1	4.09	5.2	Severe
NC-HF	1	18.9	3.03	2.9	Moderate
	2	-	-	-	CFRP bond failure
NC-MG	1	17.9	2.11	2.7	Moderate
	2	27.6	1.44	5.5	Severe

## Data Availability

The data presented in this study are available upon request from the corresponding author.

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
