# Peer review of "Building-Information-Modeling Based Approach to Simulate Strategic Location of Shelter in Place and Its Strengthening Method"

_materials, 2021, doi:10.3390/ma14133456_

Round 1

Reviewer 1 Report

BIM approach is adopted, in the submitted manuscript, to manage a shelter in place by utilizing existing facilities. How geometry charateristics, such as stair raisers, tread, and width influence the SIP evacuation time is investigated. The case study of a light railway station is considered.

Finally, the impact resistance of four different-in-type strengthened reinforced concrete walls is investigated. However, conclusions from the analysis of the results of the impact tests  appears to be quite trivial.

The paper is quite well-written, it turns out to be readable and clear, but its scientific contents are poor. This is why, the authors should better explain what are the main novelties and results of general nature from the present study.

In the followng, some remarks are listed:

Section 2.1.1: "Gaits speeds under  20", please, rephrase this sentence to clarify or, at least, add the unit of measurement.

Figure 3: please, add some comments to describe the images, both in the text and in the caption.

Figure 5: Please, rephrase the caption to render it more exhaustive.

Table 3: Please, rephrase the caption.

Table 5: please, rephrase the caption.

Figure 7: I think that the blue lines represent metal grids. Please, add a label in the image to clarify. Furthermore, rephrase the caption.

Section 4.1: "As shown in Table 6 ... to improve this case". Plaease, rephrase these sentences to clarify this crucial point.

Section 4.1.1: "In addition, rooms 95 and 135 ... in a narrow area". Please, rephrase this sentence and better describe Figure 8a to clarify this point.

Section 4.1.2 Please, clarify that the same conclusions for Scenarios A and B are reached.

Section 4.1.3. It is not clear why the authors claim that the people concentration in Room 135 is solved quickly. Please, better explain this point.

Section 4.2.1: "This study ....for each case". This part should be moved above, before section 4.2.1.

Table 7: Please, rephrase the caption.

Section 4.2.6: Please, "increased" should be substituted by "increasing".

Section 4.3.1: Please, "Table 7" should be substituted by "Table 8".

Section 4.3.1: "For all psychological ... displacement ratio" Please, rephrase this sentece to clarify this point.

Section 4.3.1. It is not correct to talk about crack distribution. So, please substitute "Crack distribution" with "crack scenario" (even in the caption of Figure 10).

Author Response

BIM approach is adopted, in the submitted manuscript, to manage a shelter in place by utilizing existing facilities. How geometry charateristics, such as stair raisers, tread, and width influence the SIP evacuation time is investigated. The case study of a light railway station is considered.

Finally, the impact resistance of four different-in-type strengthened reinforced concrete walls is investigated. However, conclusions from the analysis of the results of the impact tests appears to be quite trivial.

The paper is quite well-written, it turns out to be readable and clear, but its scientific contents are poor. This is why, the authors should better explain what are the main novelties and results of general nature from the present study.

In the following, some remarks are listed:

Answer: First of all, thank you very much for your useful comments on our paper. We have carefully considered all your comments, and the revised manuscript is now attached for your reconsideration. We really appreciate the opportunity to resubmit. Also, we would like to thank you for your excellent comments which significantly improved the quality of our paper.

1) Section 2.1.1: "Gaits speeds under 20", please, rephrase this sentence to clarify or, at least, add the unit of measurement.

Answer: As you recommended, the information is modified.

2) Figure 3: please, add some comments to describe the images, both in the text and in the caption.

Answer: As you recommended, the information is modified.

3) Figure 5: Please, rephrase the caption to render it more exhaustive.

Answer: As you recommended, the information is modified.

4) Table 3: Please, rephrase the caption.

Answer: As you recommended, the information is modified.

5) Table 5: please, rephrase the caption.

Answer: As you recommended, the information is modified.

6) Figure 7: I think that the blue lines represent metal grids. Please, add a label in the image to clarify. Furthermore, rephrase the caption.

Answer: As you recommended, the information is modified.

7) Section 4.1: "As shown in Table 6 ... to improve this case". Plaease, rephrase these sentences to clarify this crucial point.

 Answer: Thank you for your recommendation.

It is erased the sentence you mentioned and supplemented it as follows.

 “As shown in Table 6, it is assumed that occupants need to be accommodated for less than 24 hours where facilities to sleep are not required in the CBRE (chemical, biological, radiological, and high-yield explosive) situation. Evacuations of the third floor and that of second and third floors were considered in the Scenarios A and B, respectively. The designation the third floor as the SIP showed that the accessibility was low and that the distance to the SIP was too large, resulting in a long evacuation time. Therefore, this study suggests and reviews Scenarios C and D to improve this case.

In FEMA 453 (May 2006) [19], the space required for SIP is divided into the case of staying less than 24 hours and the case of staying more than 24 hours. In this study, the minimum area of SIP was defined into the case of staying less than 24 hours and all scenarios are simulated. As shown in Table 6, both the case where the shelter location is located on the 3rd basement (Scenario A) and the case where it is divided into 2nd and 3rd basement (Scenario B) were considered. If the third basement is designated as SIP, the distance to the shelter will be too far and the evacuation time will be longer. In this study, in order to improve this, it is simulated considering the case of installing the stair only the upward line (Scenario C) and the case of installing it both the upward and the downward line (Scenario D).”

8) Section 4.1.1: "In addition, rooms 95 and 135 ... in a narrow area". Please, rephrase this sentence and better describe Figure 8a to clarify this point.

Answer: Thank you for your recommendation.

It is modified it as follows.

“Scenario A designated the third basement as the SIP and set all occupants to move to the third basement during CBRE circumstance. The simulation results showed that the minimum time needed by occupants to reach the third basement was 47.4 s, the maximum time was 396.6 s, and the average time was 218.7 s. The minimum, maximum, and average distances traveled by occupants were 14.0, 189.3, and 105.9 m, respectively. In addition, rooms 95 and 135, which had to be pass after 20 s to go to the emergency stairs, had bottlenecks due to overlapping occupants in a narrow area (Figure 8 (a)). As shown in Figure 8 (b), the bottleneck is sufficiently severe for room135 to use up to 10 people in a small area of 4.626 m2 in the simulation time of 146–162 s. There is a risk of another accident during the evacuation. In addition, after 20 seconds elapsed during the simulation, people gathered intensively to the emergency stairs going down to the 3rd basement which is designated the SIP location in this scenario from the 1st basement. There was a bottleneck in which occupants overlapped in rooms 95 and 135 refer to Figure 8 (a). As follow Figure 8 (b) when 150 seconds of simulation time elapsed, it was confirmed that a serious bottleneck occurred as a maximum of 10 people gathered in Room 135.”

9) Section 4.1.2 Please, clarify that the same conclusions for Scenarios A and B are reached.

Answer: Thank you for your recommendation.

It is added following comment to compare scenario A and B.

“The minimum, maximum, and average distances traveled by occupants were 15.2m, 178.1m, and 90.6m, respectively. Like Scenario A, in Scenario B, bottlenecks occurred in Room96 and Room135 for moving to the emergency stairs, but as the maximum distance required for evacuation was shortened by 5.92% from 189.3m to 178.1m, the time taken to evacuate from 397.5sec to 390.0sec. Improved by 1.66%.”

10) Section 4.1.3. It is not clear why the authors claim that the people concentration in Room 135 is solved quickly. Please, better explain this point.

Answer:

In the case of scenarios A and B, all 213 occupants were evacuated from the first basement level to the second basement level or the second basement level and the third basement level, which are designated the SIP, so the bottleneck occurred as all occupants were concentrated in Room 95 and Room 135. However, in scenarios C and D, the number of people evacuated from the first basement floor (121 occupants) and those coming up from the fourth basement (92 occupants) were distributed, reducing the bottleneck in Rooms 95 and 135.

11) Section 4.2.1: "This study ....for each case". This part should be moved above, before section 4.2.1.

Answer: As you recommended, the information is modified.

12) Table 7: Please, rephrase the caption.

Answer: As you recommended, the information is modified.

13) Section 4.2.6: Please, "increased" should be substituted by "increasing".

Answer: As you recommended, the information is modified.

14) Section 4.3.1: Please, "Table 7" should be substituted by "Table 8".

Answer: As you recommended, the information is modified.

15) Section 4.3.1: "For all psychological ... displacement ratio" Please, rephrase this sentece to clarify this point.

Answer: Thank you for your recommendation. The sentence is deleted, which is a mistake statement.

16) Section 4.3.1. It is not correct to talk about crack distribution. So, please substitute "Crack distribution" with "crack scenario" (even in the caption of Figure 10).

Answer: As you recommended, the information is modified.

Reviewer 2 Report

The manuscript discusses a very interesting subject. Is also a well-written and well-organized article. In the attach document are the reviewer's comments.

Author Response

Answer: First of all, thank you very much for your useful comments on our paper. We have carefully considered all your comments, and the revised manuscript is now attached for your reconsideration. We really appreciate the opportunity to resubmit. Also, we would like to thank you for your excellent comments which significantly improved the quality of our paper.

1) Page 1, Abstract

Answer: As you recommended, the information is modified.

2) Figure 1, put the legend of both lines.

Answer: As you recommended, the information is modified.

3) Table 1, is not clear what is S1, S2, …. and K.

Answer: As you recommended, the information is modified.

S denotes the speed along the line of travel, K denotes constant, as shown in Table 1.

4) Figure 3, the legend of figure must describe the (a) and (b).

Answer: As you recommended, the information is modified.

5) Put pathfinder or Pathfinder but equal in all text.

Answer: As you recommended, the information is modified.

6) Page 6, remove it.

Answer: As you recommended, the information is modified.

7) Table 4, The author should indicate the standa2rds follow in these test2s. 2The2y also should indicate the respective variation coefficient.

Answer: As you recommended, the information is modified.

8) Table 4, why the authors have two different tests for tensile strength? Why not the splitting test for both?.

Answer: Thank you for your recommendation. NC specimens without fibers, and HSDC with fibers, thus NC using splitting tensile strength test according to ASTM C496. And HSDC was using direct tensile strength test (dog-bone test) according to JSCE [a]. For the dog-bone shaped specimen, the cross-section of specimen was 30 mm x25 mm and length was 330 mm. Thus, it cannot test specimen with coarse aggregate (such as NC).

9) Title of Table 5..

Answer: Thank you for your recommendation, the information is modified.

10) Title of Figure 7.

Answer: Thank you for your recommendation, the information is modified.

11) Page 9. Case A, The authors must explain the meaning of these acronyms. They appear for the first time here in the text.

Answer: As you recommended, the information is modified.

The stair raiser is reduced to 180 mm considering the maximum misstep [23] and limited to the BRE (Building Research Establishment), IBC (International Building Code), and LSC (NFPA’s Life Safety Code). Cases

12) Page 9. Case A, These case (E and F) appear only here for the first time.

Answer: Thank you for your recommendation. The detail information of case E and case F was described in Table 7.

13) 4.3.1, may be a figure to show the test?.

Answer: Thank you for your recommendation. Table 8, Figure 11 and 12 shows test results in details.

14) Page 11, Table 7, correct number of table.

Answer: As you recommended, the information is modified..

15) Table 8, the authors must explain this, in the table or in the text.

Answer: Thank you for your recommendation, the information is modified. The TM 5-1300 was superseded by UFC 3-340-02.

16) Figure 10, the figures are not clear, please put also images if the test or of the spcimens, exolain also what the green color means.

Answer: Thank you for your recommendation, the information is modified (green color). The image by camera looks askew, thus, we draw it. If necessary we will provide image, but it will not clear (beautiful).

17) Page 13, Table.12 change to 2.

Answer: Thank you for your recommendation, the information is modified

Round 2

Reviewer 1 Report

The overall quality and, in particular, readability of new version of the manuscript are improved, but its scientific contents are still poor. 

Conclusions from the results of the impact tests do not add anything to the knowhow about this topic, while the results of the application to the light railway station does not appear to be of general value, although they are certeinly of interest and well explained.  

This is why I am not enthusiastic about pubblication of the submitted manuscript in Materials.

Author Response

The overall quality and, in particular, the readability of the new version of the manuscript is improved, but its scientific contents are still poor. 

Conclusions from the results of the impact tests do not add anything to the know-how about this topic, while the results of the application to the light railway station do not appear to be of general value, although they are certainly of interest and well explained.  

This is why I am not enthusiastic about publication of the submitted manuscript in Materials.

Answer: First of all, thank you very much for your useful comments on our paper. We have carefully considered all your comments, and the revised manuscript is now attached for your reconsideration. We really appreciate the opportunity to resubmit. Also, we would like to thank you for your excellent comments which significantly improved the quality of our paper.

This paper focuses on the building information modeling-based approach to simulate the strategic location of shelter in place and its strengthening method. That simulates the strategic location of SIP and suggests modification method makes up the vast important part of the paper. Furthermore, evaluate the SIP strengthening method and suggest it. The important thing is that HSDC was new materials and developed by our team, which exhibited the best strengthening materials in this paper. HSDC has great tensile properties (tensile strength and strain capacity), energy-absorbing capacity [29], and bond strength with concrete [22] (this sentence was added in the paper).

The tensile strength of the HSDC was approximately 33.0-39.2% lower than that of typical UHPC composites, which use different lengths of high-strength straight SF and 1.5% or 2.0% fiber volume content curing with heat treatment. However, the tensile strain capacity and toughness of HSDC were 2.8-4.4 times, and 0.60-0.76 higher than those of typical UHPC composites, respectively [1,2].

The UHPC hybrid using PE (polyethylene fiber, 1.5 vol.%) and SF (steel fiber, 0.5 vol.%) as the report by Kim et al. (UHPC-PE+SF) [3], showed a compressive strength of 131.0 MPa, the tensile strength of 12.1 MPa, and tensile strain capacity of approximately 2.9%. Although HSDC has lower strength and tensile toughness compared with UHPC-PE+SF, their tensile strain capacities are similar. The hybridization of PE and SF in cementitious composites has a maximum limit of fiber volume fraction, in which the tensile strain capacity stops increasing with the increase of the PE content. It is necessary to adopt a fiber type and fiber volume fraction, such as a short high-strength PE or a long high-strength SF, and to precisely obtain the right hybrid ratio of the PE and SF in cementitious composites. The poor dispersion of fibers in cementitious composites during mixing is due to the excessive amount of PE.

Furthermore, compared with typical ECC composites [4], the HSDC showed approximately 63.3-76.0% higher tensile strength, 53.7-54.5% lower tensile strain capacity, and 0.24-0.32 lower toughness. Overall, the HSDC without heat treatment exhibited perfect tensile properties compared with those of previous mixtures.

And bond strength of HSDC to concrete was evaluated with slant shear tests, conducted according to ASTM C882. The specimens with HSDC conformed to the properties (14-21MPa for 28 days) of the ACI Committee 546 recommendation. And six reinforcement concrete beams (strengthening by HSDC) were prepared and tested according to a three-point bending test, which was evaluated the flexural capacity of normal concrete beams strengthened with HSDC.

But as a matter of fact, all of the above properties were already published in Ref. 22 and 29. Therefore, the damage level, failure mode, and reaction force of the wall with three strengthening methods were described in this paper. According to the result, the specimen strengthened with HSDC exhibited great impact resistance and simply mentioned the reason in this paper using Ref. [22, 29]. This paper just describes impact test results as necessary in order to avoid redundancy with other published papers.

Therefore, hoped you can positively reconsider this constitute.

Thank you for your recommendation again. If you have any additional comments to add, be sure to let us know. We would like to thank you for your excellent comments which significantly improved the quality of our paper.  

  1. J.I. Choi, K.T. Koh, B.Y. Lee, Tensile behavior of ultra-high performance concrete according to combination of fibers, J. Korea Institute for Struct. Mainte. Insp. 19 (4) (2015) 49-56.
  2. S.H. Park, D.J. Kim, G.S. Ryu, K.T. Koh, Tensile behavior of ultra-high performance hybrid fiber reinforced concrete, Cem. Concr. Comp. 34 (2012) 172-184.
  3. M.J. Kim, D.Y. Yoo, Y.S. Yoon, Effects of geometry and hybrid ratio of steel and polyethylene fibers on the mechanical performance of ultra-high performance fiber-reinforced cementitious composites, J. Mater. Resear. Techno. 8 (2) (2019) 1835-1848.
  4. V.C. Li, S.X. Wang, C. Wu, Tensile strain-hardening behavior of polyvinyl alcohol engineered cementitious composite (PVA-ECC), ACI Mater. J. 98 (6) (2001) 483-492.
  5. Yuan, T.F.; Hong, S.H.; Shin, H.O.; Yoon, Y.S. Bond strength and flexural capacity of normal concrete beams strengthened with no-slump high-strength, high-ductility concrete. Mater. 2020, 13, 4218 (1-16).
  6. Yuan, T.F.; Lee, J.Y.; Yoon, Y.S. Enhancing the tensile capacity of no-slump high strength high ductility concrete. Cem. Con. Comp. 2020, 106, 103458 (1-10).